# Mirrors for Pacific Islander Children: Teaching Resilience Through Culturally Adapted Bibliotherapy

**DOI:** 10.3390/ijerph22030430

**Published:** 2025-03-14

**Authors:** Isabel Medina Hull, Elizabeth A. Cutrer-Párraga, Paul H. Ricks, G. E. Kawika Allen, Kendra M. Hall-Kenyon, Lorena Seu, Kristofer J. Urbina, Melia Fonoimoana Garrett

**Affiliations:** 1Alpine School District, American Fork, UT 84003, USA; isabelmhull@gmail.com; 2Counseling Psychology & Special Education, Brigham Young University, Provo, UT 84602, USA; gekawika_allen@byu.edu (G.E.K.A.); kristofer.j.urbina@gmail.com (K.J.U.); 3Teacher Education, Brigham Young University, Provo, UT 84602, USA; paul_ricks@byu.edu (P.H.R.); kendra_hall@byu.edu (K.M.H.-K.); 4Connections—The Learning Resource, Pago Pago, AS 96799, USA; lorenaseu@gmail.com; 5Nebo School District, Spanish Fork, UT 84660, USA; melia.fonoimoana@gmail.com

**Keywords:** bibliotherapy, culturally relevant education, resilience, mental health education of Pacific Islander children, American Samoa

## Abstract

Pacific Islander youth face disproportionately high rates of suicide and mental health challenges, yet culturally appropriate interventions remain scarce. This study investigated whether culturally adapted bibliotherapy could effectively promote resilience in American Samoan children by incorporating culturally relevant stories and coping strategies. Through collaboration with on-island Samoan cultural brokers, we developed and implemented culturally adapted stories and lessons designed to resonate with the experiences of children in American Samoa. The study employed a mixed-methods approach with 34 American Samoan children aged 5–13 years, using observations, adapted card sorts, and forced-choice procedures to measure responses to the intervention. The results demonstrated that participants showed strong identification with the culturally adapted stories and characters, actively engaged with the embedded coping strategies, and reported increased confidence in applying resilience techniques. These findings suggest that culturally adapted bibliotherapy holds promise as an effective method for promoting resilience in Pacific Islander children while highlighting the importance of cultural authenticity in mental health interventions.

## 1. Introduction

Compared to other countries in the Western Pacific Region, the Pacific Islands have some of the highest rates of suicide and suicide attempts, with rates peaking in the adolescent years [1]. Subica and Wu found that Pacific Islanders (PIs) are among other multiracial adolescents that “are disproportionately burdened by illicit substance use, depressed mood, and suicidality” [2] (para. 4). Suicide accounts for over 10% of deaths in Native Hawaiian and Pacific Islander males and females from the ages of 1 to 19 [3,4]. These high rates of suicide indicate significant adolescent psychosocial distress [5,6].

Although suicide rates vary across different PI communities [1,7], American Samoa has seen a particularly troubling increase [8] among youth, calling attention to the urgent need for effective mental health interventions. A survey conducted by Empowering Pacific Islander Communities with 1125 high school students in American Samoa revealed that one in three adolescents seriously considers suicide [9]. Recent studies underscore both the pressing mental health challenges and the resilience within the American Samoan community. For example, a qualitative study by Mew et al. utilizing the Pacific-specific Fa’afaletui research framework, highlighted that while American Samoa has developed “an impressive amount of infrastructure and community mobilization” to support mental health, significant challenges persist in adequately meeting the mental health needs of its youth [10]. According to the study, resources remain limited due to factors such as insufficient funding, a shortage of trained mental health professionals, and systemic barriers in integrating culturally appropriate practices into mental health service delivery. These challenges foreground the necessity for culturally relevant and targeted interventions to bridge critical gaps and ensure effective mental health care for American Samoan youth [10]. Despite the alarming rates of suicide and suicide attempts among American Samoan adolescents, research addressing the mental health of PI youth remains scarce, and studies specifically focusing on American Samoan children are virtually nonexistent. This gap in the research indicates an urgent need for targeted studies to better understand and address the mental health challenges faced by this population [11].

### 1.1. Barriers to Mental Health Care in American Samoa

The rich cultural traditions and strong community values of American Samoans provide many protective factors for well-being, including robust family support systems and spiritual practices. However, certain cultural perspectives about mental health may present unique challenges for those seeking support. Research by Cutrer-Párraga et al. and Taylor and Kuo indicates that some within these communities view mental illness as primarily affecting White individuals, potentially due to the historical dominance of Western approaches to mental health care [12,13]. This perspective can intertwine with traditional beliefs that living in accordance with cultural values and maintaining strong family and spiritual connections should be sufficient for emotional well-being [12]. While these cultural strengths indeed contribute to mental wellness, such beliefs may inadvertently create barriers for those who need additional mental health support, making it more complex for PIs to recognize and seek treatment for mental health challenges while remaining true to their cultural identity [14]. Access to mental health care, including utilization, retention, and the availability of providers and programs, continues to be a pressing issue [15]. This aligns with other research that indicates that PIs have notably lower rates of mental health service utilization and treatment retention despite experiencing elevated levels of psychological distress and facing significant barriers to accessing adequate care [11,16].

Although U.S. health systems have established mental health resources in the Pacific, colonial mental health legislation has left some regions, including American Samoa, with inadequate protection for the rights of individuals with mental illness [17]. This historical reliance on U.S. funding has resulted in infrastructure that is often limited and inconsistently supported, compounding American Samoa’s unique challenges in delivering effective mental health care [17]. Additionally, a critical shortage of qualified mental health professionals in the region further restricts the availability and consistency of services [10]. Cultural understandings, community perspectives, and stigma surrounding mental health also create significant barriers to establishing comprehensive care. These interconnected challenges emphasize the importance of developing culturally aligned mental health support systems that resonate with community values, foster long-term engagement, and ensure sustainable care tailored to the needs of American Samoan youth.

### 1.2. Building Protective Factors Through Social and Emotional Learning (SEL)

Given the elevated mental health risks and low utilization of mental health services among PI populations [11,18], SEL could be a valuable, culturally sensitive approach to improve mental health outcomes for American Samoan children. SEL frameworks, such as those developed by the Collaborative for Academic, Social, and Emotional Learning (CASEL), focus on developing self-awareness, self-management, social awareness, relationship skills, and responsible decision-making [19,20,21,22]. Research shows that SEL programs have broad benefits, including improvements in emotional well-being, academic performance, and social skills while reducing symptoms of anxiety and depression [23,24,25]. These competencies foster emotional regulation, interpersonal skills, and cognitive flexibility, all of which have consistently been associated within resilience research as having protective effects [26].

Resilience, understood as the ability to adapt and thrive through adversity, is essential for improving mental health outcomes in children [27,28,29]. More recent resilience models use a dynamic, multi-systems approach, which can be applied to systems such as families, schools, and individuals [26]. SEL skills—such as recognizing and managing emotions, seeking support, and making constructive choices—target processes that may bolster resilience factors on an individual and community levels [26,30,31]. Although there are many other processes and systems that contribute to resilience, integrating SEL into schools and community programs can strengthen children’s ability to navigate challenges and support their mental well-being in culturally relevant ways.

### 1.3. The Need for Culturally Adapted Interventions

Although the importance of SEL has been clearly demonstrated and SEL programs have been shown to benefit children of diverse backgrounds, studies and interventions that are tailored to specific cultural contexts, such as those for PI children, are scarce [24,32]. Research on trauma among PIs suggests that interventions tailored to cultural contexts can better address unmet needs and enhance social and emotional well-being [SEL] [33].

For American Samoan children, culturally adapted SEL is vital as it aligns with their unique values and leverages community-based strengths. By emphasizing protective factors such as emotional awareness, cognitive flexibility, and goal-oriented action, bibliotherapy can empower children to handle life’s challenges and enhance emotional well-being [34,35].

Thus, to improve the mental wellness of American Samoan children, mental health interventions should be culturally affirming, sensitive, and appropriate. Mental health services are most effective when they align with children’s cultural values, particularly in historically marginalized communities [36]. However, existing research often overlooks the distinct cultural needs of PIs, which can lead to overlooking the distinct identities and cultural values of these children. Terms like “ethnic glossing” [37] and “ethnic lumping” [16] were coined by researchers noticing a problematic and overgeneralized approach that fails to address the unique identities and values of American Samoan children. Effective SEL interventions should focus on culturally specific protective factors that reinforce both individual strengths and community support. For example, Ungar emphasizes resilience as not only an individual’s ability to adapt but also the community’s capacity to provide meaningful support [35]. Additionally, culturally adapted interventions should acknowledge collective and historical traumas that have impacted the mental wellness of Indigenous and PI communities [33,38].

Studies show that culturally sensitive resilience programs can significantly improve mental health outcomes. For instance, the Community Resiliency Model (CRM) in Sierra Leone and the “Wear Your Pride” program for Samoan American youth demonstrated that culturally adapted resilience training can reduce stress and improve social support, leadership, and cultural pride [39,40]. Furthermore, Jongen et al. found that resilience interventions for Indigenous adolescents led to enhanced coping skills, social connections, and mental health outcomes [41].

The integration of cultural values into resilience-building efforts can create meaningful, long-lasting improvements in mental wellness for American Samoan children. By strengthening both individual resilience assets and multi-system supports, culturally tailored interventions offer a promising approach to address mental health disparities in these communities [26,35].

### 1.4. Bibliotherapy as a Culturally Adapted Mental Health Intervention

Bibliotherapy has a rich history, dating back to times when stories were shared around campfires, providing guidance and imparting wisdom [42]. This longstanding tradition shapes an intrinsic link between ancient storytelling practices and modern bibliotherapy, illustrating how narratives have long been used as tools for emotional and psychological healing. In ancient civilizations, storytelling and poetry played a vital role in religious rites aimed at enhancing individual and community well-being [43,44]. The healing power of literature was recognized in Ancient Greece, where a library bore the inscription, “the healing place of the soul” [45] (p. 1). Figures like Aristotle used readings to evoke emotional healing, while Cornelius Celsus recommended reading as a therapeutic tool for the ill to aid in decision-making [42,46]. In Samoan culture, the tradition of *talanoa*—a form of open and respectful storytelling—has been central to conveying wisdom, values, and emotional healing across generations. This oral tradition provided a culturally relevant framework for well-being, serving both to educate and to create spaces for emotional expression and communal connection [47,48].

Over the centuries, bibliotherapy evolved to encompass a range of definitions and practices. Lundsteen described bibliotherapy as “getting the right book to the right child at the right time about the right problem” [49] (p. 505). Others have categorized bibliotherapy based on the genre, setting, or medium used [50]. Suvilehto et al. described bibliotherapy as “the therapeutic use of stories and connected activities to help young children cope with social and emotional problems” [44] (p. 300). Despite varied definitions, bibliotherapy is generally understood as the intentional use of story literature to support therapeutic goals [42,51,52].

Research has shown that bibliotherapy can enhance children’s problem-solving skills, increase their sense of resourcefulness, and support them in coping with trauma [51]. For example, bibliotherapy has been applied effectively with children who have experienced traumatic events, including natural disasters [53], familial suicide [54,55,56], and depression [57,58,59]. In a study involving children in residential care, bibliotherapy significantly reduced social anxiety and decreased behavior problems [60]. In addition to reducing behavioral and emotional issues, bibliotherapy can teach essential life skills. Everall et al. found that through bibliotherapy, children learn cognitive restructuring techniques that empower them to view life challenges as manageable and to choose their responses [30]. When children relate to characters in the stories, they are more likely to internalize these skills and apply them to their own lives.

#### 1.4.1. Stages of Bibliotherapy

For bibliotherapy to succeed, children must see themselves reflected in the characters and narratives. The first stage of bibliotherapy, identification, relies on this connection [42,61]. For PI children, seeing characters who reflect their own experiences, values, and cultural practices can foster a stronger sense of connection to the story, thereby enhancing the therapeutic process [62]. Stewart and Ames spotlight the importance of selecting culturally and contextually relevant materials in making bibliotherapy effective for children [63]. Their study found that using culturally responsive literature not only supported children’s ability to identify with characters but also enhanced empathy, tolerance, and a sense of belonging.

Once children identify with characters, they move through catharsis, insight, and universalization. In the catharsis stage, children experience an emotional release by connecting with characters facing similar challenges [61]. Through this shared journey, they begin to process and understand their own feelings. This is followed by insight, where children gain new perspectives on their situations by seeing how characters navigate similar issues [64]. Finally, in the universalization stage, children realize they are not alone, finding comfort in shared human experiences [52,63].

#### 1.4.2. Implementing Bibliotherapy

The process of implementing bibliotherapy involves several steps. First, interventionists should assess each child’s unique experiences, selecting stories that align with their needs and background [61,65]. Next, the chosen book is read aloud, with pauses for discussion, allowing children to reflect on the story and relate it to their own lives [65]. Finally, children engage in a post-reading activity that reinforces the story’s lessons, helping them process the material through creative expression [42].

#### 1.4.3. Culturally Adapted Bibliotherapy

By providing reflections and images of students’ cultures through books, adults can enhance students’ self-esteem, engagement, social and emotional coping skills, and academic performance [62,66]. According to Gay, there are “strong correlations between culturally responsive teaching and the school achievement of students of color” [67] (p. 627). To provide more culturally responsive literature in Guam, Jackson and Heath created stories that reflected critical features of the Chamorro culture and were modeled after “nursery rhymes, fairy tales, and local indigenous stories and legends” [62] (p. 465). Although the stories were not written in the Chamorro language, in attempts to be inclusive of students and teachers that did not speak the language, the stories were written to strengthen Chamorro identity. Additionally, some of the literature addressed social–emotional wellness topics, such as self-esteem.

Stoicovy found that the oral and written retelling of Chamorro legends provided motivation and scaffolds for literary acquisition for four Chamorro children [68]. In addition to the academic benefits, the use of cultural legends also provided opportunities for students to experience “connectedness as family members and as a culture” [68] (p. 157). In another study on the use of retellings of Chamorro legends, the performance of one fifth-grade student diagnosed with ADHD significantly improved [69]. Although the student had a history of disruptive behaviors and of struggling in writing, during all retelling sessions, the student remained on task and demonstrated comprehension, linguistic spillover, and story structure, suggesting that the use of culturally adapted resources was more well suited to fit his needs [69].

Culturally affirming bibliotherapy has also been used to help African American students cope with trauma after being displaced due to Hurricane Katrina [63]. These students participated in reading circles with books that were chosen for their positive reflections of the cultural heritage of the students involved and topics around family changes or unfortunate situations. Additionally, the books contained universal themes such as friendship and self-confidence to increase protective resilience factors. As the students connected with the characters and the situations they faced, they began articulating their own feelings of loss and isolation. Additionally, “they were able to utilize the strategies employed by those characters to address the problems they faced in their own lives” [63] (p. 233). In providing students with culturally affirming and thematically appropriate bibliotherapy, students were able to build resilience factors at an individual level [63].

Standard book-sharing practices often fail to align with the lived realities of diverse children, particularly in communities rich with cultural heritage and unique perspectives. For culturally responsive bibliotherapy to succeed, it must build on students’ everyday experiences by incorporating cultural details such as legends, religion, and relationships [66,70]. The diverse cultures of the Pacific Islands, for instance, hold rich histories and storytelling traditions [71] that are integral to their identity yet are seldom represented in children’s literature [67]. Without such authentic representation, standard book-sharing approaches risk becoming increasingly irrelevant “in the real world that never was and never will be all white” [72] (para. 16).

Even among the books listed in the American Library Association’s Asian/Pacific American Award for Literature, most texts portray characters from Asia, especially from Japan and China, and those that do contain PI characters often focused on people from Hawaii. It is insufficient to focus only on social and emotional learning without also considering the cultural backgrounds of the children being taught [67]. Following a systematic process for culturally adapting stories can provide students with resources that are more compatible with their needs and values.

## 2. Study Purpose

Although data on the mental health of PI youth remain limited, existing evidence suggests the importance of culturally appropriate and accessible mental health resources for American Samoan children. High rates of adolescent suicide, depression, and substance abuse indicate the challenges these children face, rooted in systemic and historical factors such as colonization, which disrupted traditional systems of well-being [2,5,6,73]. While the United States has introduced some mental health resources, many remain inadequate, inaccessible, or misaligned with the cultural context of American Samoa [1,17]. Strengthening mental health resources that build upon existing cultural values and promote coping strategies within community frameworks can provide American Samoan children with tools to navigate adversity and enhance well-being [74]. This approach acknowledges the strengths inherent in the culture while addressing gaps in support systems such as building emotional and mental health.

The purpose of this study was to develop culturally adapted mental health resources that equip American Samoan children with social and emotional skills tailored to their cultural context. By embedding SEL within a culturally relevant framework of bibliotherapy, we aimed to empower these children with culturally affirming strategies to manage mental and emotional challenges, thereby reducing the risk of self-destructive behaviors and suicide [31]. This approach was supported by research indicating that positive mental health can moderate suicidal ideation and decrease the likelihood of such thoughts translating into actions [75].

## 3. Method

### 3.1. Research Design

This exploratory study investigated identification—the critical first step of bibliotherapy—among PI children engaging with culturally adapted stories. Identification, which precedes catharsis, insight, and universalization in bibliotherapy, involves connecting with characters’ ethnicities, values, and cultural traditions [61]. Given the limited research on culturally adapted literature for PI youth, we examined American Samoan children’s identification with characters and perspectives on culturally adapted books through guided readings, discussions, and post-reading activities.

A qualitative methodology was selected to explore participants’ experiences and perspectives, particularly the “what” and “how” of their engagement with the texts. Data collection included card sorts, forced-choice procedures, and field notes, allowing researchers to examine social phenomena through observation and interpretation of participants’ lived experiences in naturalistic settings [76,77].

### 3.2. Study Context and Researcher Trustworthiness

#### 3.2.1. Context

Our research originated when Samoan research team members highlighted the adolescent suicide crisis in American Samoa among adolescents [9]. Given the expertise of our research team with resilience and bibliotherapy, this prompted our investigation into potential preventive interventions, focusing on building resilience factors through bibliotherapy as a possible mitigator of future mental health challenges. Following preliminary research on these approaches, we conducted the study during a summer reading program on island at a public library.

To address potential subjective biases, we incorporated reflexivity throughout the data analysis process [78]. The bidirectional influence between the primary researcher’s background and data analysis is examined in detail in Section 9. Further, our research team practiced reflexivity [78] by explicitly acknowledging our Western epistemological perspectives and actively seeking cultural guidance. Beginning one year before on-island interventions, we established regular video conferences with American Samoan professionals who served as cultural brokers. These consultations were instrumental in developing culturally adapted stories and lesson plans, creating culturally relevant assessment measures, understanding local cultural attitudes and practices, and anticipating potential cultural barriers.

During the study’s implementation on island, we collaborated with Samoan translators and library staff who provided essential cultural and linguistic support. They helped bridge communication gaps by clarifying concepts for children, advising on effective communication strategies, including speech patterns, facilitating culturally appropriate student engagement and behavior management, and providing real-time cultural context and interpretation.

#### 3.2.2. Participants

Participants’ ages ranged from 5 to 13 years with a mean of 8.5. Participants’ grade level ranged from pre-school to seventh grade. A majority of participants were Samoan (82%) and female (56%). Some participants were more fluent in Samoan than English, although all participants understood at least some English. During the entirety of our time with the participants, at least one library worker was available to help with interpretation. Participants were not excluded based on socioeconomic status, gender, or race/ethnicity. There was a total of 34 participants across five days. On day 5, when data collection measures were completed, 31 participants were present. Participant demographics are depicted below in Table 1.

### 3.3. Recruitment

Participant recruitment in American Samoa employed multiple culturally appropriate strategies, including distributing flyers to parent organizations, church groups, and community centers, as well as broadcasting announcements on local radio stations, which leveraged the oral traditions of Samoan culture. Snowball sampling, recognized for its effectiveness in community-based and potentially vulnerable populations [79], was also used. To ensure broad community engagement, we collaborated with our cultural brokers on island, local educational organizations, and the public library to disseminate information. Recruitment materials and informed consent procedures provided detailed explanations of the study’s purpose, procedures, and potential benefits to the American Samoan community. Confidentiality and privacy were given and explained to all participants and their parents. They were also informed about the time their information will be stored, people who will have access to the record, and the expiration date of the records.

## 4. Intervention Procedures

The intervention phase consisted of structured SEL lessons. These lessons were followed by bibliotherapy sessions using culturally adapted literature. All intervention components were developed through extensive consultation with American Samoan cultural brokers to ensure cultural appropriateness and relevance.

### 4.1. SEL Lessons

Initial consultations with American Samoan experts revealed limited emotional vocabulary within the culture, particularly regarding three commonly experienced emotions: sadness, anger, and fear. This insight informed our foundational lessons on emotional identification and understanding, with subsequent sessions focusing specifically on these three emotions.

### 4.2. Culturally Adapted Stories

Prior to the intervention, our team developed and preliminarily published eight children’s books adapted to American Samoan culture. Our team identified story themes through interviews with on-island educators who recommended including both male and female characters, addressing grief and loss, exploring bullying, particularly physical confrontations, and incorporating the cultural legend of Turtle and Shark as recurring characters. Table 2 illustrates the alignment of emotional themes and main characters across the books. Complete story summaries are provided in Appendix A.

### 4.3. Bibliotherapy Implementation

Following the SEL lessons, the bibliotherapy sessions implemented a four-step bibliotherapy protocol including book selection, interactive read-alouds, post-reading discussion, and a problem-solving activity. As part of the overall interaction with the child participants, our cultural brokers emphasized incorporating song, movement, and color elements, opening safety affirmations (“In this space you are safe. We will never scream at you or hit you. You are safe here.”), a whole group setting for the SEL lessons, and small group configurations for the interactive read-alouds. Table 3 outlines the emotional focus and corresponding books for each session. Figure 1 documents story read-alouds used in the bibliotherapy sessions. A sample lesson plan and materials are provided in Appendix B.

## 5. Data Collection

Data were collected regarding American Samoan children’s perceptions of culturally adapted stories about bullying and grieving in a variety of ways including observations, field notes, forced-choice procedures, and adapted card shorts.

### 5.1. Observations and Field Notes

To optimize observational data collection while maintaining authentic story engagement, participants were divided into small groups for interactive read-aloud sessions. Each group was staffed by two research team members: a primary reader presenting culturally adapted stories and a dedicated observer documenting field notes. This dual-role structure facilitated the systematic documentation of verbal responses, emotional reactions, and behavioral engagement. Field notes, fundamental to rigorous qualitative research since the early 1900s, are particularly crucial in descriptive cultural studies [80,81]. Following best practices, our observers recorded detailed notes immediately after each session, ensuring contextual documentation without audio or video recording, thereby preserving cultural sensitivity and participant comfort [81]. To maintain confidentiality, participants were assigned pseudonyms, with physical data secured in locked cabinets and digital data stored in password-protected databases.

### 5.2. Forced-Choice Rankings

To evaluate story preferences and inform future refinement of culturally adapted literature for American Samoan children, we implemented forced-choice ranking procedures. Forced-choice ranking methods have been effectively utilized in research involving children to assess preferences and attitudes. For instance, a study by Wetzel et al. compared the effectiveness of multidimensional forced-choice formats to traditional rating scales in preventing faking among children [82]. Their findings suggest that forced-choice formats can provide more accurate assessments of children’s preferences. In this study, participants were required to create hierarchical rankings of all stories, eliminating the possibility of tied responses and providing clear preference data. The procedure was administered collectively following the completion of all reading sessions. To facilitate the ranking process, all culturally adapted stories were displayed simultaneously on the floor, allowing participants full visual access to the materials. Two research team members provided literacy support, assisting participants with title identification and written responses when needed. This accommodating approach ensured that reading and writing abilities did not confound participants’ ability to express their story preferences. However, given London et al.’s recommendation that research with young children should also incorporate other forms of data collection methods in addition to force-choice items [83], we supplemented the forced-choice rankings with an adapted card sort procedure.

### 5.3. Adapted Card Sorts

Following the SEL lessons and read-aloud sessions, we conducted an adapted card sort to assess participants’ experiences with the first stage of bibliotherapy (identification), their responses to cultural representation in stories, and their suggestions for story modifications. The methodology was specifically adapted to align with cultural learning preferences in American Samoa, as identified through consultation with cultural brokers.

Traditional card sorts employ either open methods, where participants create their own categorization systems, or closed methods, utilizing predetermined categories [84]. These procedures often incorporate facilitators to moderate discussions and mitigate potential group dynamic issues [12,85,86]. However, consultation with our cultural brokers taught us that American Samoan children typically engage in passive learning styles and defer to elders. This is in contrast with Western constructivist approaches. To align with these norms, we adapted traditional card sort methods, enabling meaningful participation while respecting cultural practices.

The adapted card sort was administered in small groups of three to five children, a size determined through cultural broker consultation to be most culturally appropriate. Each group was facilitated by one research team member who presented a poster board displaying three to four predetermined response categories per card. Participants used sticky notes to indicate their responses to each card presented, with an “agree”, “disagree”, or “I have a comment” category to encourage additional feedback. To ensure authentic responses, facilitators emphasized that the purpose was to gather honest feedback about the books, there were no incorrect answers, and negative feedback would not cause offense. Table 4 details the complete card set and corresponding response categories, while Figure 2 illustrates the physical setup of the card sort activity.

To help ensure children felt at ease and minimize power dynamics, other adults were excluded from data collection sessions. This included local in-culture adults. The sessions were facilitated by a team of four individuals: three female and one male. Among them, three were students at the time of the study—one a graduate student in school psychology, one a graduate student in special education, and one an undergraduate student in special education. The fourth facilitator was a PhD-level researcher specializing in SEL and bibliotherapy. Facilitators identified as Latina/o/x (*n =* 2), Native Hawaiian or Other Pacific Islander (NHOPI, *n =* 1), and White (*n =* 1). To respect cultural norms and encourage authentic responses from the children, the research team opted not to use video or audio recordings. Instead, facilitators took detailed field notes and photographed the card sort arrangements for subsequent analysis. Additional annotations were added after each session to provide contextual insights into the discussions that took place within each group. 

## 6. Data Analysis

Because this study explored research questions that had not been extensively investigated within this population, thematic analysis was chosen to analyze the data. According to Braun and Clarke, “Thematic analysis is a method for identifying, analyzing and reporting patterns (themes) within data” [87] (p. 79). To ensure rigor, the six phases of reflexive thematic analysis were followed as a guide.

The process began with familiarizing ourselves with the data by reading it multiple times. While no formal coding occurred during this initial phase, observations and analytic memos were recorded [88]. Once the familiarization phase was complete, we created codes to capture prevalent topics discussed by participants, such as families, animals, and clothing. Each code was tagged and named within the text, with prevalence defined as any individual occurrence of the topic across all data sets. To ensure no data were overlooked, the data sets were revisited multiple times.

Following initial coding, we systematically organized the coded data extracts into thematic categories. Each code served as a category header, under which relevant data extracts were collated into structured bullet-point lists. This organizational approach facilitated the identification of patterns and relationships across the dataset while maintaining the context of individual extracts. Through analysis of the data extracts within each code, patterns began to emerge. For instance, under the code “clothing”, a recurring theme of immodesty in character illustrations was identified when clothing was mentioned. Throughout the process of developing, reviewing, and refining themes, we cross-referenced data extracts from both the collated codes and the raw data to identify supporting or contradictory evidence. Ultimately, thematic analysis revealed themes and patterns centered on children’s experiences with the books, the books’ shortcomings, and their appropriateness. The final analysis incorporated participant quotes, excerpts, and illustrations from the books for support and illustration.

In addition to thematic analysis, participants ranked the books from most to least favorite. First-place rankings received one point, while rankings in eighth place received eight points. Each book’s total score was calculated by summing all its assigned points. Lower total scores indicated a higher preference for the book. This ranking system provided quantitative data to complement the thematic analysis.

### Trustworthiness

We attended to trustworthiness throughout our study by implementing multiple strategies [89]. According to Lincoln and Guba, trustworthiness in qualitative research encompasses four key criteria: credibility (confidence in the truth of the findings), transferability (the applicability of findings to other contexts), dependability (the consistency and replicability of findings), and confirmability (the degree of neutrality or researcher bias mitigation) [90].

To establish credibility, we employed methodological triangulation through multiple data collection measures and conducted peer debriefing sessions to explore additional analytical perspectives [89]. We maintained strict adherence to the research plan, documenting any deviations and their justifications. Our analysis included examination of deviant cases that challenged primary findings [91], strengthening the reliability of our conclusions.

To support transferability, we provided “thick” detailed descriptions of the research context and participants [92]. The dependability of our findings was enhanced using multiple analysts examining data for themes and categories. For confirmability, we implemented researcher reflexivity through documented self-reflection on data interactions [78]. Further strengthening our trustworthiness, we engaged three Samoan expert reviewers [93] to evaluate our results for plausibility. These reviewers, who resided on-island and possessed expertise in social emotional learning, represented diverse academic backgrounds (psychotherapy, education, and anthropology), self-identified genders (one male and two females), and educational attainment levels (one bachelor’s and two graduate degrees). Their feedback was incorporated into the final findings [78], enhancing both the cultural validity and overall trustworthiness of the study.

## 7. Findings

### 7.1. Book-Ranking Results

The analysis of forced-choice rankings revealed that books addressing grief and loss resonated most strongly with participants. The most preferred book, *Sefina*, follows a young girl processing her mother’s death. When feeling isolated in her grief, she encounters Turtle and Shark—characters from Samoan legend—who help her recognize existing support systems and encourage her to reconnect with friends, particularly her classmate Lani. The second-most preferred book, *Makoa and His Scary Feelings*, similarly addresses parental loss, following a boy coping with his father’s death. In this narrative, Makoa seeks solitude at his father’s favorite fishing spot before Turtle and Shark guide him toward accepting familial support, ultimately leading to a healing conversation with his mother. Notably, both preferred narratives incorporate culturally significant elements while addressing grief experiences through an accessible, hope-centered lens.

In contrast, *Maleko’s Adventures: Troubles at School*, which explores bullying through the story of a young merman helping his friend Kai confront a bully named Fiafia, was consistently ranked least preferred. Despite incorporating the cultural motifs of Turtle and Shark and addressing the culturally relevant theme of bullying, the fantasy elements and resolution (discovering the bully’s loneliness and forming friendship) appeared less engaging to participants. Complete rankings are presented in Table 5, with detailed book summaries available in Appendix A.

### 7.2. Participants’ Perspectives

The analysis of the adapted card sort data revealed consistent themes across participant groups regarding story elements. Participants evaluated multiple aspects of the narratives, including visual components (illustration colors, animal characters, and character clothing) and narrative elements (character relationships and conflict resolution). The field notes and observational data corroborated these emergent themes. Participants provided detailed feedback within each category, identifying both successful elements and areas for potential enhancement.

### 7.3. Illustrations

Participants provided insightful feedback on the visual elements of the culturally adapted stories. While they appreciated the overall creativity and beauty of the illustrations, specific cultural and aesthetic preferences emerged.

In terms of color choices, despite the varied palette used throughout the books, participants expressed a clear preference for brighter, more vibrant colors over the pastel tones prevalent in many illustrations (see Figure 3). They noted that pastel colors are generally unpopular in American Samoa. Several illustrations were described as “too dark” (see Figure 4), with participants consistently requesting more luminous color choices.

Participants identified several culturally authentic details in the illustrations, particularly appreciating the accurate depiction of sandals placed outside home entryways and the vibrant representations of mountains and beaches. These elements resonated with their lived experience of the island environment. However, to increase cultural authenticity, participants recommended including illustrations of local marketplaces, which they identified as a significant yet missing feature of island life. They also suggested technical improvements, specifically requesting higher image quality and larger text size for better readability.

### 7.4. Clothing

Cultural modesty expectations emerged as a significant theme during both interactive read-alouds and card sort discussions. Participants consistently emphasized the importance of appropriate clothing choices in the illustrations, particularly highlighting the absence of traditional garments like ‘ie-lavalavas (traditional cloth worn around the waist). This feedback, which reflected deep-rooted cultural values regarding modest dress, was unexpected given the careful attention to making the books culturally appropriate. Despite these efforts, the omission of culturally significant clothing called attention to the implicit expectations surrounding cultural representation in visual elements. Specific concerns centered on illustrations depicted what participants perceived as insufficient clothing coverage. Comments such as “her clothes are too little for her” and “you should make the clothes more decent” highlighted their discomfort especially with illustrations showing exposed stomachs (see Figure 5). This feedback extended across various clothing types, including shorts, shirts, and bathing suits that participants felt did not align with American Samoan modesty standards.

### 7.5. Animals

The inclusion of animal characters, particularly Turtle and Shark from Samoan legend, emerged as a significant theme in participant feedback. Student participants consistently expressed appreciation for these cultural icons appearing throughout all eight books, with several participants specifically noting their enjoyment of the various artistic interpretations of these characters across the series (see Appendix A for illustrations of Turtle and Shark in each book).

While participants overwhelmingly supported the inclusion of Turtle and Shark, they also suggested expanding the range of animal representation. Participants recommended increasing the presence of fish and other sea animals alongside Turtle and Shark, while others advocated for incorporating additional animals familiar to island life, such as goats, dogs, and butterflies. This feedback suggests that while culturally significant animals resonated strongly with participants, they also desired representation of the broader ecological diversity present in their daily island experience.

### 7.6. Characters

The analysis of participant responses revealed strong themes around social connections and character relationships in the stories. While participants appreciated the existing friendships portrayed, they consistently advocated for expanding the character roster to better reflect American Samoan social dynamics.

Their recommendations emphasized the cultural value of community and connection, specifically calling for more balanced representation of male and female characters, increased presence of extended family members (including infants), and additional peer relationships. Participants also suggested incorporating more social activities, particularly those involving food and communal gathering, reflecting the centrality of shared meals in American Samoan culture.

The cultural significance of avoiding isolation emerged strongly in participant comments, with children making statements like, “Add more characters so no one is alone, and everyone is important” and “no one is alone here”. These observations reflected a cultural discomfort with depictions of solitary characters, such as in Figure 6, where the main character appears alone. Interestingly, this finding stood out given the multiple meetings we held with our cultural brokers who did not mention this aspect. It could be that the value of community is so deeply ingrained in the culture that our cultural brokers did not consciously think to highlight it. In contrast, the children, seeing the isolated characters in the illustrations and hearing the stories read aloud, noticed the lack of cultural alignment with their lived experiences. This contrast between the perspectives of the children and the cultural brokers helped our team better understand the importance of including diverse voices—including children—when assessing cultural appropriateness.

Participants also offered specific critiques regarding character authenticity, particularly focusing on naming conventions and character types. A notable example emerged with the character named “Fiafia”, which elicited laughter during read-alouds as participants explained that “fiafia” in Samoan means “to like”, highlighting the need for more culturally appropriate character names.

Dialogue emerged as another area for improvement, with several participants indicating that characters needed more extensive verbal interaction. The most significant character critique centered on the inclusion of a merman character (depicted in Figure 7), which created cultural disconnection. Participants expressed confusion about the concept of a merman, requiring explanation of the term, and students explicitly stated that the merman character did not reflect anyone in their real-life experiences.

However, this critique of the fantastical merman character stood in marked contrast to participants’ responses to other characters in the stories. Most participants indicated that these other characters successfully reflected people from their daily lives, suggesting that grounding characters in familiar cultural contexts resonates more effectively with American Samoan children than introducing fantasy elements outside their cultural framework.

### 7.7. Problems and Solutions

Participants responded positively to the books’ approach to problem-solving and emotional coping strategies. They particularly valued narratives that offered practical solutions to relatable challenges, with participants noting how the stories reflected their own experiences and potential coping strategies. Another participant highlighted the moral dimension, appreciating “how Maleko learned to help others be kind”.

The stories’ emphasis on seeking support resonated strongly with participants, who valued messages like “if there is trouble, I can go to others and ask for help”. This theme of community support was particularly evident in the two highest-ranked books, *Sefina* and *Makoa and His Scary Feelings*, both addressing parental loss through the lens of family and community support.

While participants generally approved of characters’ decision-making, they identified instances where actions conflicted with cultural values. Notably, they criticized a scene depicting a character yelling at their teacher (Figure 8), emphasizing that such behavior violates the strong cultural value of respecting elders in American Samoa. Overall, however, reception was notably positive, with participants describing the books as “fun and make you smile” and expressing interest in personal copies and future readings.

### 7.8. Identification

The analysis of card sort responses also demonstrated strong participant identification with story elements, a critical initial phase of bibliotherapy. Participants disclosed they connected with both character traits and narrative challenges, recognizing themselves and community members in the stories’ depictions.

### 7.9. Identifying with Characters

Participants connected with story characters by recognizing familiar physical and behavioral traits, which mirrored their daily interactions with community members. They connected characters to various people in their lives, including immediate family members (parents and siblings), extended family (aunties and uncles), and friends. The cultural authenticity of character representation was particularly evident when several participants specifically noted that Leilani, the protagonist of “Leilani Learns to Love”, appeared authentically American Samoan.

Personal connections also emerged spontaneously during interactive reading sessions. During a reading of *Maleko’s Adventures: Troubles at School*, students immediately recognized physical similarities between the main character and one of their peers, noting specific details down to hairstyle. Character identification extended beyond physical appearance, however. One participant observed that while some characters might not physically resemble people they knew, the characters’ behaviors and actions authentically reflected their community members. This feedback suggests that, although there were some minor issues raised by participants, the stories succeeded in capturing both visual and behavioral aspects of American Samoan culture overall.

### 7.10. Identifying with Problems and Solutions

Participants demonstrated strong emotional engagement with the stories, often sharing personal experiences that paralleled the narratives’ themes. During discussions of bullying-themed stories, several participants disclosed their own experiences with bullying, both at school and within family contexts. The stories addressing loss particularly resonated with one participant sharing their experience of losing their mother at a young age, while another discussed their 9-year-old cousin’s similar loss. Notably, this participant emphasized how the stories provided practical guidance for supporting their grieving cousin. Several other participants stated that they learned what to do when loved ones pass away.

During the card sort, when presented with the card “I learned something”, many participants shared different possible solutions they learned to help them during challenging times, which seemed to indicate participants’ internalization of coping strategies. Their insights reflected deep engagement with the stories’ themes and solutions. “In life there may be problems but there are also solutions, like talking to someone”, one participant observed, while another noted, “I learned that even though we have hard times we shouldn’t run away from them”. These responses demonstrated an understanding of both emotional challenges and constructive responses.

Participants identified multiple coping strategies from the stories and accompanying bibliotherapy lessons, including seeking support from parents and other adults, embracing peer support (“kids could help kids”), understanding social consequences (linking bullying to social isolation), and applying specific calming techniques, particularly “belly breathing”. The prevalence of solution-focused responses during the card sort suggests that participants not only connected with the stories’ challenges but also internalized practical coping strategies.

## 8. Expert Reviewers’ Responses

The roles of the three expert reviewers were distinct from those of our cultural brokers. The cultural brokers played a critical role in the preparatory stages of the study by helping us design the research, understand the cultural context, and develop culturally appropriate methods for collecting trustworthy data. For example, they contributed to the adaptation of card sorts, as described in the Section 3, ensuring that our approach was aligned with local practices with children.

In contrast, the three expert reviewers were tasked with reviewing the findings after the data collection to assess their plausibility based on their expertise and to provide feedback on unexpected or noteworthy results. This process aligns with robust qualitative data analysis practices aimed at ensuring trustworthiness as triangulating perspectives from cultural insiders (brokers) and external expert reviewers enhances the credibility of the findings [94].

The expert reviewers confirmed the overall plausibility of the findings but highlighted two surprising results. First, they noted children’s strong preference for stories addressing grief and loss, which contrasted with the cultural taboo surrounding discussions of death with children in American Samoa. Second, they found it unexpected that the children did not comment on the body sizes of the characters in the stories, even though these did not reflect the typical body types commonly found on the island. These insights from the expert reviewers provided valuable reflections on the findings, enriching our understanding and enhancing the study’s trustworthiness. The reviewers also proposed several enhancements to increase cultural authenticity. The visual representations could better reflect local demographics through more diverse body sizes. They emphasized incorporating culturally significant elements of grieving practices, particularly the exchange of sacred mats. To strengthen community recognition, they recommended including iconic cultural features such as the village “fale” (traditional meeting house) and the *Sa* bell. The Sa bell holds particular cultural significance in American Samoan daily life, marking a communal period of prayer and reflection each evening. The Sa period is announced by three bells: the first rings five minutes before it begins, the second signals the start of the Sa, and the third marks its end. Typically starting around 6 PM and lasting between 7 and 15 min, this time prompts a pause in community activities—children return home, traffic halts, and businesses temporarily close. The bell, often a repurposed oxygen tank, serves as an auditory anchor for this sacred tradition, embodying the collective rhythm and values of island life.

Consulting expert reviewers aligned with Polynesian cultural traditions where it is common to seek counsel from elders or matriarchs, reflect on their wisdom, and later report back to them. Therefore, engaging with expert reviewers was both culturally sensitive and affirming, honoring these values within the research process [95].

## 9. Discussion

This study investigated how American Samoan children engaged with culturally adapted stories, focusing on the critical first phase of bibliotherapy: identification. The results indicated that participants successfully identified with characters who reflected familiar physical traits, behaviors, and challenges from their community. Their feedback on both successful elements and areas for improvement offered four clear directions for strengthening cultural resonance in future story adaptations, advancing the development of culturally appropriate SEL resources for American Samoan children.

### 9.1. Direction One: Role of Language

The findings from this study align with previous research emphasizing the critical role of language in cultural adaptation of interventions [62,96,97]. Specifically, they support assertions that effective cultural adaptation requires both accurate language use and a deep understanding of cultural relevance [96,97]. Participants’ responses demonstrated this linguistic sensitivity: while they responded positively to the inclusion of Samoan language in the stories, they quickly identified culturally incongruent elements. For example, they critiqued the use of “Fiafia” as a character name, recognizing it as a misuse of a Samoan word meaning “to like”. Similarly, the introduction of unfamiliar concepts like “merman” required explanation, highlighting a disconnect from cultural understanding. These linguistic and conceptual misalignments were particularly evident in *Maleko’s Adventures: Troubles at School*, which received the lowest ranking among the stories, suggesting that appropriate language use influences cultural resonance.

### 9.2. Direction Two: Alignment with Cultural Values

Alignment with cultural values, which goes beyond the inclusion of indigenous language, is required to ensure cultural resonance. For example, *Always in Your Heart* was the only bilingual (English/Samoan) story in our collection. Participants shared initial enthusiasm for the Samoan language elements during guided reading; however, the story ranked fifth overall. This suggests that language inclusion, while important, is insufficient for creating meaningful cultural connections.

The story’s limited appeal appears to stem from misalignment with fundamental American Samoan cultural values. Specifically, the illustrations depicted characters in clothing considered immodest by community standards, and the plot included a child yelling at their teacher—behavior that violates the deeply held cultural value of elder respect. These departures from cultural norms likely undermined the potential benefits of bilingual presentation.

Participants connected strongly with authentic cultural details, from customs like removing shoes before entering homes to depictions of extended family living and familiar island landscapes. Their suggestions for enhancement—including local marketplaces and brighter colors reflecting island aesthetics—emphasized the role of accurate cultural representation in engaging young readers.

### 9.3. Direction Three: Alignment with Cultural Contexts

Our findings also support research emphasizing that bibliotherapeutic solutions must align with cultural contexts [96,97]. The highest-ranked stories, *Sefina* and *Makoa and His Scary Feelings*, featured coping strategies centered on family support, reflecting Polynesian collectivistic values [36,98]. However, during card sort activities, participants also readily identified coping strategies from the stories, suggesting progression beyond initial identification into the later bibliotherapy phases of insight and universalization. Similar to Stewart and Ames’s findings with African American students post-Hurricane Katrina, our participants gained problem-solving skills through culturally affirming bibliotherapy [63]. This indicates that when coping strategies align with cultural values, bibliotherapy can effectively support children’s emotional development.

While our findings confirm the importance of collectivistic solutions for PI children, they also reveal the value of teaching individual coping strategies. During card sort activities, participants identified both communal support systems and personal regulation techniques, particularly culturally adapted breathing exercises. For example, a traditional calming technique involving “breathing in” the smell of hot chocolate was modified to reference “Koko Samoa”, a traditional Samoan beverage, making the strategy culturally relevant while maintaining its therapeutic value. This dual approach—combining culturally aligned collective support with adapted individual coping strategies—appeared to provide the participants with a set of social and emotional tools, equipping them for situations whether alone or within their community.

### 9.4. Direction Four: Use of Realistic Problems and Solutions

Findings from the current study support previous research emphasizing the importance of realistic problems and solutions in culturally adapted literature [65]. The stories addressed challenges common to American Samoan children—particularly bullying and the loss of loved ones—enabling strong reader identification. Notably, the two highest-ranked books, *Sefina* and *Makoa and His Scary Feelings*, both addressed parental loss, suggesting that authentic representation of difficult experiences resonated deeply with participants.

While our study primarily focused on identification, the first phase of bibliotherapy, participants frequently shared personal experiences of bullying and loss during both guided reading and card sort activities. This spontaneous sharing suggests participants also experienced catharsis, the second phase of bibliotherapy, indicating that culturally authentic problem representation can facilitate multiple stages of the bibliotherapeutic process.

## 10. Implications

Culturally authentic representation can enhance children’s engagement with resilience-building literature, potentially mitigating future mental health challenges in PI children and youth. Our findings suggest four key elements for effective cultural adaptation, described above. Misalignment with language, cultural content, or visual representations can disrupt reader engagement and diminish therapeutic value. For PI populations, effective SEL resources should fulfill the following:Emphasize community support and harmonious problem-solving;Depict extended family and friendship networks;Include culturally relevant animal characters;Provide both collective and individual coping mechanisms.

This comprehensive approach to cultural adaptation enhances bibliotherapy’s potential for building social and emotional skills, potentially serving as a preventive measure against mental health challenges while promoting overall emotional wellness. By seeing their cultural values and experiences reflected in literature, children can more readily internalize adaptive coping strategies and resilience factors.

### 10.1. Limitations and Future Research

There are several limitations to this study. The primary limitation of this study is that the research team and the creators of the books were comprised of cultural outsiders. Because of this, it was imperative that the research team be sensitive to relationships and tensions that exist between outsiders and insiders [99]. As Bernal et al. and Lee et al. both outline, one must acknowledge the way ethnic similarities and differences shape relationships and the intervention [96,97]. The adaptations were completed with the guidance of our cultural brokers. While Turtle and Shark were included as characters in every story as a reference to a well-known cultural legend, the research team was careful not to appropriate the legend.

Despite our commitment to learning about the culture and our long-term collaboration with cultural insiders, the culturally adapted books were still “imperfect” mirrors. There were some errors in the representation of both the physical and social environment, as was evident in the feedback provided by participants. One of these imperfections was that the illustrations did not accurately represent the typical body size in American Samoa. Although it was not mentioned by any child, this imperfection was brought to our attention by the expert reviewers. It is possible that the children did feel comfortable talking about the body size despite the measures taken to create a safe space. It is also possible that they were too focused on the clothing that could be considered immodest to focus on the size of the characters.

Spending time on the island further educated our research team on insider knowledge we did not possess, and mistakes made in the books; however, future research should ensure cultural insiders play a more active role in the creation and editing process of the books. This is also important considering that in 2019, only two books included at least one PI creator [100].

The feedback provided by participants should also be interpreted cautiously. Although we had a total of 34 participants, it is important to remember that there is still much within-culture diversity and no one source of feedback should serve as authority over what is and is not culturally authentic for any cultural group.

While this study examined how students were able to identify with the books and their experiences relative to identification, it did not specifically examine the participants experiences with the other stages of bibliotherapy. Although there was evidence that the participants experienced those stages, we did not analyze the stages of catharsis, insight, and universalization, which should be further examined in future studies to understand the experiences of participants more fully.

Additionally, while this study primarily focused on books that served as mirrors, allowing children to see themselves and their experiences reflected, we now acknowledge the importance of exploring stories that act as windows into other cultures. Exposure to narratives from diverse backgrounds may provide children with valuable insights and new ways of thinking that they might not otherwise encounter. Additionally, we recognize the potential of using stories that create a certain distance—serving as deflectors—to facilitate emotional processing and broaden perspectives. Future studies should further investigate how books from non-dominant cultures can support both identification and transformation in children’s social–emotional development.

### 10.2. Conclusion: Books as Mirrors

Overall, the findings from this study reinforce Bishop’s metaphor of books as mirrors, emphasizing the importance of cultural representation in children’s literature [101]. This is particularly significant given that only 0.4% of children’s books in 2022 included PI content [102]. Like the experiences documented by Guevara and Perez, our participants demonstrated strong engagement when encountering reflections of their lived experiences in the literature [103,104]. Participants readily identified with characters and recognized familiar cultural environments. For many, this represented their first experience seeing themselves reflected in the literature. Their emotional engagement extended to connecting with characters’ experiences of bullying and loss, particularly the cultural emphasis on family support systems. This deep identification, achieved through collaboration with cultural insiders, proved crucial for the success of bibliotherapy [63], enabling participants to engage with and internalize protective factors presented in the stories. The study demonstrates how culturally authentic representation can create meaningful literary mirrors for populations traditionally underrepresented in children’s literature.

This study investigated how American Samoan children engaged with culturally adapted stories, specifically examining their identification with characters and situations—a critical first step in bibliotherapy. While bibliotherapy has proven effective for teaching social and emotional skills, its application within PI populations remained unexplored. Given the extreme scarcity of children’s literature featuring PI content and social-emotional themes (less than 0.4%) [102], our team developed culturally adapted stories with guidance from cultural insiders.

The findings revealed several key elements that facilitated participant engagement. Environmental representation through familiar landscapes and cultural icons helped create authentic connections. Character authenticity, including physical appearance and family dynamics, enabled personal identification. Linguistic accuracy proved crucial for cultural resonance. The emphasis on friendship and family relationships aligned with fundamental cultural values.

Although some story elements required refinement, participants successfully identified with multiple aspects of cultural representation. This identification process enabled engagement with therapeutic elements, suggesting that culturally adapted bibliotherapy could serve as an effective, accessible tool for promoting social and emotional well-being in American Samoan children. These findings provide a framework for developing culturally authentic mental health resources that resonate with and support PI youth.

## Figures and Tables

**Figure 1 ijerph-22-00430-f001:**
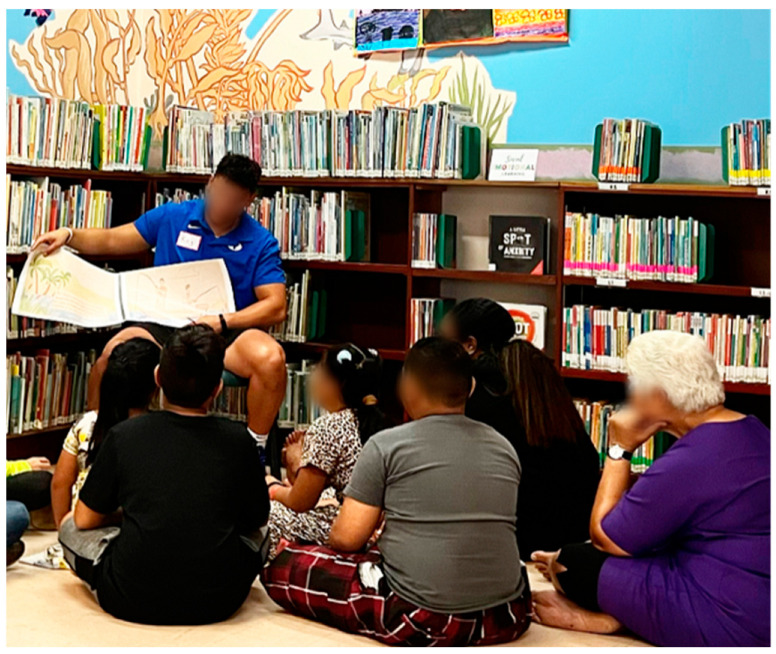
Picture of students participating in guided readings.

**Figure 2 ijerph-22-00430-f002:**
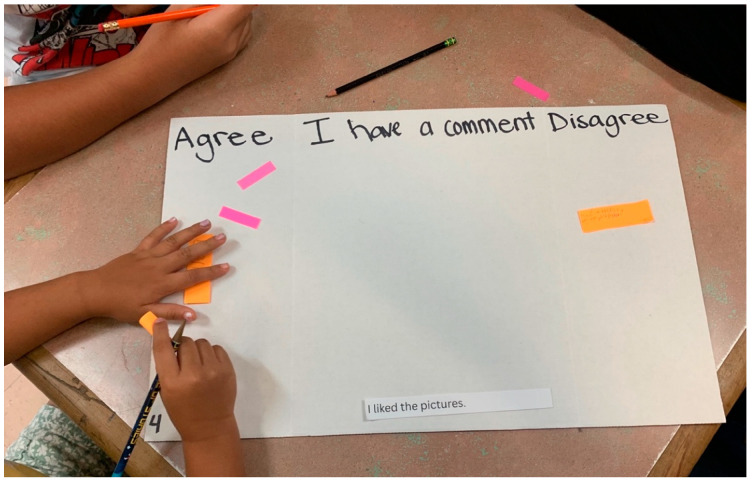
Picture of adapted card sort.

**Figure 3 ijerph-22-00430-f003:**
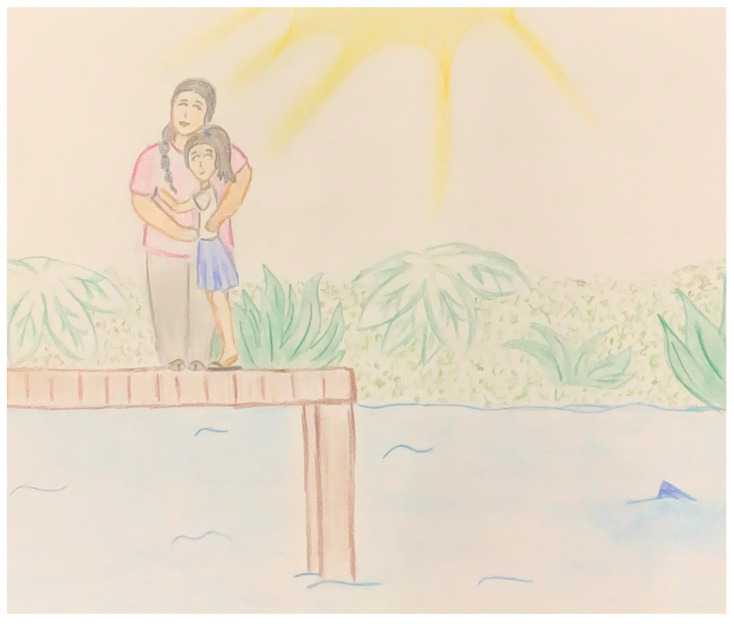
Illustration with pastel colors from *Dad’s Favorite Spot*.

**Figure 4 ijerph-22-00430-f004:**
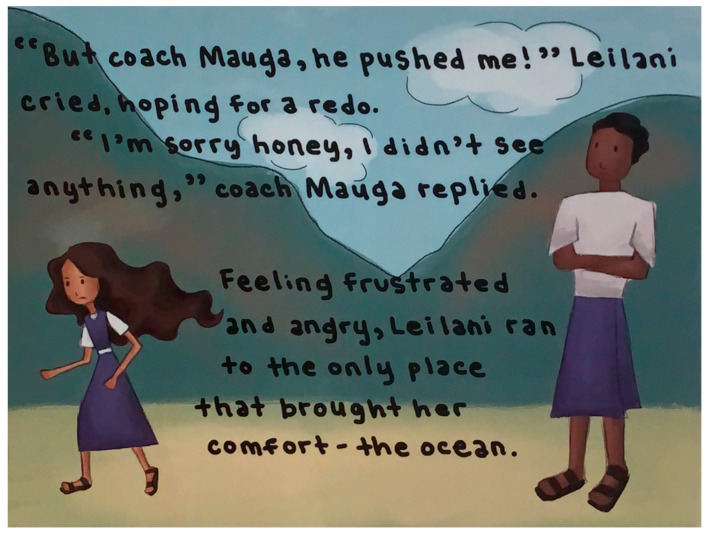
Illustration with dark colors from *Leilani Learns to Love*.

**Figure 5 ijerph-22-00430-f005:**
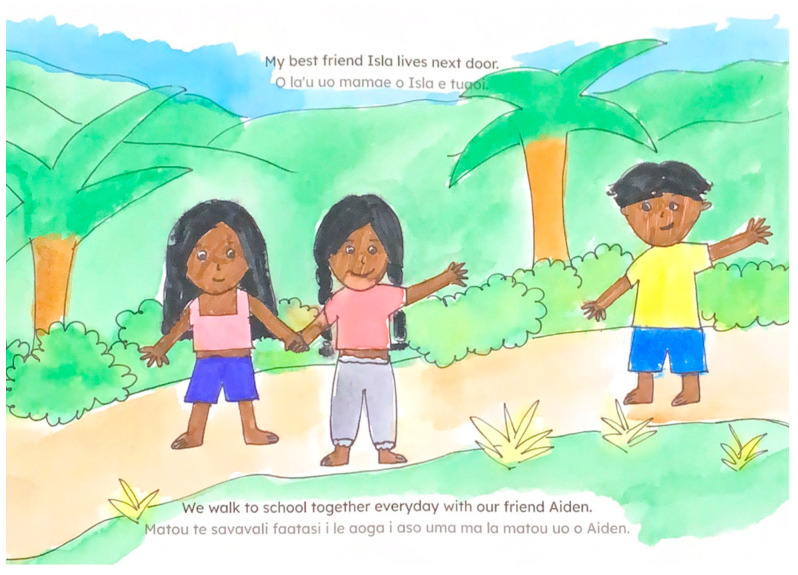
Image of immodest clothing from *Always in Your Heart*.

**Figure 6 ijerph-22-00430-f006:**
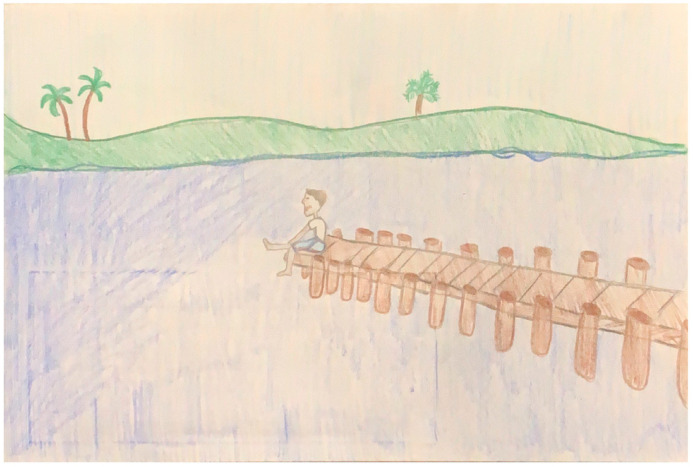
Illustration from *Makoa and His Scary Feelings*.

**Figure 7 ijerph-22-00430-f007:**
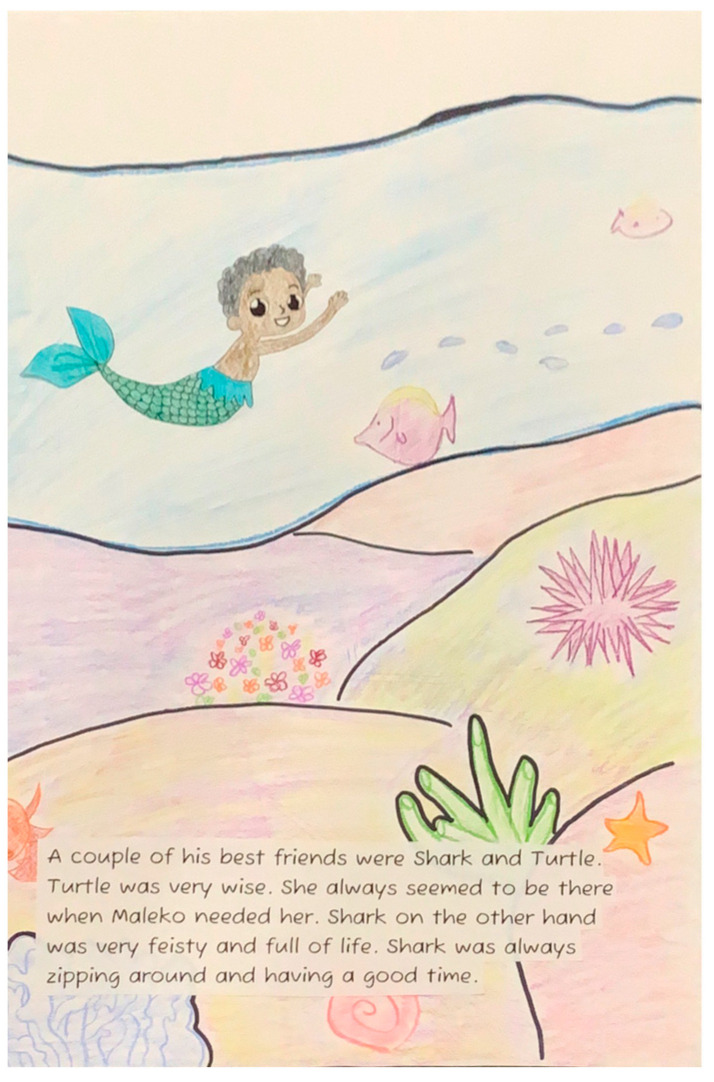
Illustration from *Maleko’s Adventures*.

**Figure 8 ijerph-22-00430-f008:**
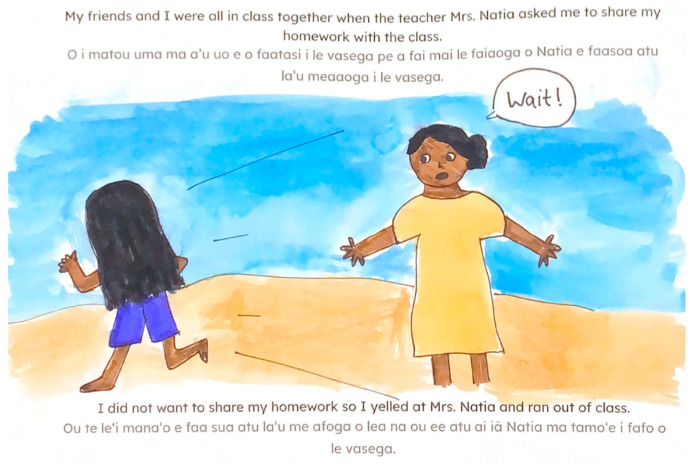
Illustration from *Always in Your Heart* of student yelling at their teacher.

**Table 1 ijerph-22-00430-t001:** Participant demographics.

Characteristic	*n*	%
Gender		
Female	19	56
Male	15	44
Age		
5	2	6.3
6	4	12.5
7	5	15.6
8	6	18.8
9	3	9.4
10	8	25
12	2	6.3
13	2	6.3
Grade level		
Pre-K	2	6.7
Kindergarten	3	10
1st	7	23.3
2nd	6	20
3rd	2	6.7
4th	5	16.7
5th	1	3.3
6th	2	6.7
7th	2	6.7
Ethnicity (participants checked all that applied)		
Samoan	28	82.4
Fijian	5	14.7
Micronesian	1	2.9
Hispanic or Latino	2	5.9
Asian	1	2.9
White	3	8.8

**Table 2 ijerph-22-00430-t002:** Book topics.

Book	Emotion	Book Topic
Book 1	Sad	Female main character dealing with loss of family member
Book 2	Mad	Female main character dealing with loss of family member
Book 3	Scared	Female main character dealing with loss of family member
Book 4	Sad	Male main character dealing with loss of family member
Book 5	Mad	Male main character dealing with loss of family member
Book 6	Scared	Male main character dealing with loss of family member
Book 7	Mad	Female main character dealing with being bullied (hit) at school by classmate
Book 8	Scared	Male main character dealing with being bullied (hit) at school by classmate

**Table 3 ijerph-22-00430-t003:** Lesson subjects.

Day	Lesson Subject	Books
Day 1	Awareness of emotions	No books were read on this day
Day 2	Identifying and expressing sadness	Dad’s Favorite Spot; Makoa and His Scary Feelings
Day 3	Identifying and expressing anger	Leilani Learns to Love; Aiden’s Anger; Always in Your Heart
Day 4	Identifying and expressing fear	Malosi Means Strength, Maleko’s Adventures: Troubles at School; Sefina
Day 5	Review of emotions and coping strategies	No books were read on this day

**Table 4 ijerph-22-00430-t004:** Card sort cards and categories.

Card	Categories
I liked the books.	Agree, Disagree, I have a comment
I liked the characters.	Agree, Disagree, I have a comment
I liked the pictures.	Agree, Disagree, I have a comment
I learned something.	Agree, Disagree, I have a comment
I would change a character.	Agree, Disagree, I have a comment
I would change the pictures.	Agree, Disagree, I have a comment
The character(s) felt like someone I would know in real life.	Agree, Disagree, I have a comment
How would you make the books better?	Change the pictures, change the characters, change the problem, Other
Who was your favorite character?	Shark, Turtle, Kids, Other

**Table 5 ijerph-22-00430-t005:** Book-ranking results.

Overall Ranking	Book Title
1st	*Sefina*
2nd	*Makoa and His Scary Feelings*
3rd	*Malosi Means Strength*
4th	*Leilani Learns to Love*
5th	*Always in Your Heart*
6th	*Dad’s Favorite Spot*
7th	*Aiden’s Anger*
8th	*Maleko’s Adventures: Troubles at School*

## Data Availability

The datasets presented in this article are not readily available due to IRB restrictions protecting the confidentiality of vulnerable participants including minors discussing sensitive mental health topics. Requests to access the datasets should be directed to Elizabeth Cutrer-Párraga at ecutrer@byu.edu, though access may be restricted based on IRB requirements and participant protection protocols.

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
