# Peer review of "Mirrors for Pacific Islander Children: Teaching Resilience Through Culturally Adapted Bibliotherapy"

_ijerph, 2025, doi:10.3390/ijerph22030430_

Round 1
Reviewer 1 Report
Comments and Suggestions for Authors
This is a beautifully written paper about a carefully constructed study of American Samoan children’s responses to a bibliotherapy activity that was meticulously culturally adapted to their community. The very important and notable contribution of this study is its’ analysis of the elements of bibliotherapy that contribute to and/or detract from its effectiveness. The authors use a meticulous qualitative analysis that gives confidence to their results. The careful attention to the American Samoan children’s response to illustration and story elements provides a very strong example of the analyses needed to create stories and books that support bibliotherapy strategies. In reading through the paper’s analysis, I found myself hoping that we might see similarly careful analyses of the children’s literature that is often used with newly arrived immigrant children, inner city communities, and children from the small, sparsely populated communities on the Great Plains.
I very much appreciate the deep respect that the paper shows for the Samoan American culture, both in the purpose of the study, the preparation of materials that might be used to strengthen the emotional regulation of Samoan-American children, and the active and ongoing participatory inclusion of community members and American Samoan professionals in conducting and understanding the study and its results.
The only jarring note in the paper is its suggestion that the culturally adapted bibliotherapy activities are ‘teaching resilience.’ The intervention seems mostly about awareness of and self-regulation of emotions – a competency that contributes to resilience but is not equivalent to resilience. While resilience indeed includes a child’s acquisition of skills to manage challenge emotions and situations, it is much more than that; resilience emerges out of social environments that incorporate large numbers of protective factors in children’s daily lives. (Masten’s (2014) Ordinary Magic). Some specific examples: (page 3) Line106 is a bit confusing in saying that social emotional learning is closely aligned with resilience, but then says that resilience-building must be integrated into SEL. Lines 116-120 appear to equate resilience training with training in components of SEL. I believe that this paper is a very valuable analysis of a bibliotherapy strategy for teaching children an important social emotional skill; and that individual mastery of emotion regulating strategies is one aspect of resilience. As one example, resilience as represented in Masten’s (2015) Ordinary Magic is much broader than an individual’s self-regulation; it also includes the creation of social environments for children that incorporate large numbers of protective factors (i.e. close bond with at least one caretaker, effective parenting, nurturing from other caring adults, moments of joy, prosocial organizations, effective schools). It would be possible to revise this paper so that it never mentioned ‘resilience’ and it would lose none of its value and relevance.
Masten, A. S. (2015) Ordinary Magic: Resilience in Development. Guilford
Reviewer 2 Report
Comments and Suggestions for Authors
This study is timely, well-researched and inspiring. I would love to use it for my bibliotherapy classes once it is published. Perhaps for further studies, a deeper look into the possibility of cultural "transference" may be meaningful as a next step. While cultural identify is an important element to draw a reader in, cultural diversity in reading materials is a way to add perspectives. I do not mean to give the children with dominant pop literature. I wonder, for example, how Anishinaabe children would respond to Samoan stories, or the vice versa. This thought is not a critique of the article; it is inspired by it.
